

# Understanding the experience of psychopathology after intimate partner violence: the role of personality

Paulo A.S. Moreira[1,2], Márcia Pinto[2,3], C. Robert Cloninger[4], Daniela Rodrigues[2,5] and Carlos Fernandes da Silva[6,7]

[1] Instituto de Psicologia e de Ciências da Educação, Universidade Lusíada—Norte (Porto), Porto, Portugal
[2] Centro de Investigação em Psicologia para o Desenvolvimento, CIPD, Porto, Portugal
[3] Centro de Acolhimento Temporário Âncora, Associação para o Desenvolvimento de Rebordosa, Rebordosa, Portugal
[4] School of Medicine, Washington University in St. Louis, St. Louis, MO, United States of America
[5] Estabelecimento Prisional de Santa Cruz do Bispo—Masculino, Direção Geral de Reinserção e Serviços Prisionais, Matosinhos, Portugal
[6] Departamento de Educação e Psicologia, Universidade de Aveiro, Aveiro, Portugal
[7] Center for Health Technology and Services Research, CINTESIS, Porto, Portugal

Corresponding author
Paulo A.S. Moreira,
paulomoreira@por.ulusiada.pt

## ABSTRACT

**Objective(s).** To fully understand the dynamics of Intimate Partner Violence (IPV) it is necessary to understand the role of personality. The current understanding of which personality characteristics are associated with IPV victimization is, however, far from comprehensive. Given this gap in the literature, our objective was to examine the associations between the dimensions of the psychobiological model of personality and psychopathological symptoms in women who had experienced IPV.

**Methods.** Using a case-control design, a group of women who had experienced IPV and who were living in shelters ($n = 50$) were compared to a group of control women who had not experienced IPV ($n = 50$). All women completed the Temperament and Character Inventory–Revised and the Brief Symptom Inventory.

**Results.** Victims of IPV showed significantly higher levels of Harm Avoidance and Self-Transcendence, and lower levels of Reward Dependence and Self-Directedness, than the non-IPV control group. Victims of IPV also reported elevated levels of psychopathological symptoms. Personality dimensions showed a broadly consistent pattern of associations across different psychopathological symptoms. A regression analysis indicated that Novelty Seeking was negatively associated with psychopathological symptoms in victims of IPV, but not significantly associated in non-victims.

**Conclusions.** The study highlights the important role of Harm Avoidance and Self-Directedness for understanding psychopathological symptoms. Novelty Seeking appears to play an important role in the expression of individuals' experiences of IPV. These results have important implications for research and practice, particularly the development and implementation of interventions.

## INTRODUCTION

Violence occurring between family members has been broadly defined as domestic violence (*Goodey, 2005*). Intimate Partner Violence (IPV) refers to physical, psychological and/or sexual violence that occurs between intimate partners (including cohabiting or divorced intimate partners, independent of gender; *Niolon et al., 2017*). IPV is a universal phenomenon (*Devries et al., 2013*; *O'Doherty et al., 2014*), and studies have revealed high rates of violence against women in particular. Worryingly, a recent survey revealed that one in five women in the European Union has been a victim of physical and/or sexual violence by an intimate partner (*European Union Agency for Fundamental Rights, 2014*).

The experience of IPV is associated with negative psychological effects (*Golding, 1999*), but the type and severity of these effects may depend somewhat on a victim's personality. The present study aimed to evaluate the associations between the dimensions of the psychobiological model of personality (*Cloninger, 1987*; *Cloninger, 1994*) and psychopathological symptoms in women who had experienced IPV. The following subsections will present past research that describes the psychological impact of IPV and the potential influence of personality, and then introduce the psychobiological model of personality.

### IPV and mental health

Exposure to IPV has a negative impact on multiple domains of an individual's biopsychosocial functioning. Research articles that directly address the associations between IPV and mental health are less common than those that describe the physical consequences of IPV (*Devries et al., 2013*; *Van Parys et al., 2014*). Nonetheless, several influential studies have considered the impact of IPV on mental health. A meta-analysis conducted by *Golding (1999)* reported that IPV has a significant influence on victims' mental health and is a strong predictor of different types of mental disorders. The negative impact of IPV on victims' psychological functioning has been confirmed more recently by two independent systematic reviews (*Dillon et al., 2013*; *Lagdon, Armour & Stringer, 2014*). Fewer studies, however, have considered how individual differences in personality dimensions serve to buffer or enhance the negative impact of IPV on mental health.

The experience of IPV is a stressor with a strong negative impact on the various biopsychological dimensions of health (*Bonomi et al., 2006*; *Coker et al., 2002*; *Dillon et al., 2013*; *Nurius & Macy, 2010*; *Renner & Whitney, 2010*; *Zlotnick, Johnson & Kohn, 2006*) and on the functioning of victims' children (*Preto & Moreira, 2012*). The high neurophysiological activation associated with abusive relationships (*Okuda et al., 2011*) impacts on victims' psychological functioning (*Jarvis, Gordon & Novaco, 2005*; *Leiner et al., 2008*) and thereby constitutes a risk factor for the development of psychopathology (*Golding, 1999*; *Hathaway et al., 2000*; *Pico-Alfonso et al., 2006*) with high levels of co-morbidity (*Lagdon, Armour & Stringer, 2014*; *Armour & Sleath, 2014*). Concordantly, the prevalence of mental disorders in victims of IPV is high (*Lagdon, Armour & Stringer, 2014*; *Okuda et al., 2011*; *Sesar, Šimic & Dodaj, 2015*). Such disorders include depression (*Beydoun et al., 2012*; *Sesar, Šimic & Dodaj, 2015*; *Stein & Kennedy, 2001*), eating disorders (*Brady, 2008*), substance abuse and addiction (*Humphreys et al., 2005*; *Simonelli, Pasquali*

& De Palo, 2014), anxiety (*Pico-Alfonso, 2005*; *Simmons et al., 2008*), and trauma related disorders (*Coker et al., 2005*). Victims of IPV also present a high prevalence of schizoid personality organizations (*Pérez-Testor et al., 2007*) and paranoid, borderline and schizotypal personality disorders (*Khan, Welch & Zilmer, 1993*; *Pico-Alfonso, Echeburúa & Martinez, 2008*). It is, therefore, of paramount importance to acquire a detailed understanding of the association between IPV and mental health in order to improve the effectiveness of clinical interventions and prognoses of victims.

## Personality and mental health

Personality is a strong predictor of mental health and a causal antecedent of mental disorders (*Cloninger, 2004*). A small body of research has documented how personality influences the relationship between IPV and psychosocial functioning. As an example, the experience of IPV has been shown to reduce the protective effect of high education on the chances of child maltreatment (*Salazar et al., 2014*). However, studies describing how personality characteristics are associated with the experience of IPV are scarce. As evidence of this, the systematic review conducted by *Lagdon, Armour & Stringer (2014)* on the effects of experiencing IPV on mental health did not include personality as a search term, and only one study included in the review described personality-related dimensions (*Sharhabani-Arzy, Amir & Swisa, 2005*).

Although all women who experience IPV are at increased risk for psychopathology, not all women develop the same symptoms. Moreover, those who present the same symptoms do not necessarily experience them with same severity (*Nurius & Macy, 2010*). To understand a victim's developmental trajectory toward resiliency/vulnerability to psychopathology after experiencing IPV it is necessary to understand the personality organizations and dynamics involved in biopsychological functioning (*Josefsson et al., 2013*). Personality dimensions are linked to differences in the risk of experiencing IPV (*Loas, Cormier & Perez-Diaz, 2011*; *Sharhabani-Arzy, Amir & Swisa, 2005*). These dimensions also play an important role in a victim's functioning, including engagement in abusive or aggressive situations (*Hellmuth & McNulty, 2008*), failing to stop abusive relationships (*Palker-Correll & Marcus, 2004*; *Renner & Slack, 2006*; *Walker, 2009*), and identifying and selecting coping strategies (*Connor-Smith & Flachsbart, 2007*). Personality dimensions also mediate the relationship between coping strategies and psychopathology (*Auerbach et al., 2010*), predict positive and negative aspects of health (*Cloninger & Zohar, 2010*; *Cloninger, Zohar & Cloninger, 2010*; *Dyrenforth et al., 2010*; *Gonçalves & Cloninger, 2010*; *Josefsson et al., 2011*), and are associated with responses to trauma, including the development and recovery of post-traumatic stress disorder (*Kunst, 2010*; *North, Abbacchi & Cloninger, 2012*). Considering this evidence, it is clear that obtaining a description of the personality organizations associated with IPV will prove useful for understanding the processes of change underlying the adaptive responses of victims (including those that are spontaneous, and those prompted by interventions).

## The psychobiological model of personality

The few studies to evaluate the personality of IPV victims have typically made use of the Minnesota Multiphasic Personality Inventory-2, the Millon Clinical Multiaxial

Inventory-I and II (MCMI-II-*Millon, 1987*; *Hart, Dutton & Newlove, 1993*), the Personality Assessment Inventory (PAI; *Morey, 1991*; *Tsang & Stanford, 2007*), and the Five-Factor Model (*Hellmuth & McNulty, 2008*). Such studies, however, are limited by the fact they tested personality models that do not facilitate an investigation of the psychobiological dynamics involved. Fortunately, the psychobiological model is an integrative approach to understanding the concept of personality as a non-linear developmental process within the individual (*Cloninger & Svrakic, 1997*; *Cloninger, Svrakic & Przybeck, 1993*; *Svrakic et al., 1993*). According to this model, personality comprises two major dimensions: temperament and character.

Temperament refers to the dispositional and stable tendencies to respond to basic emotional stimuli (danger, novelty and reward) with automatic habitual reactions, including inhibition, activation and behavioral maintenance (*Cloninger, Svrakic & Przybeck, 1993*). Four temperament dimensions remain stable throughout an individual's development. Novelty Seeking (NS) describes the tendency to respond intensely to novel stimuli as signals of pleasure. Such responses include an active approach to rewarding stimuli and active avoidance of negative stimuli. Individuals with high NS are exploratory, impulsive, and have a disliking for rules and regulations. Harm Avoidance (HA) describes the propensity to respond intensely to signals of punishment or a frustrating loss of reward by inhibiting behavior. People with high HA tend to be anxious, fearful and shy. Reward Dependence (RD) describes the tendency to respond intensely to signals of reward, especially social approval and attachment cues, and individuals with high RD are sentimental, sociable and friendly in attachments. Finally, Persistence (Ps) describes the propensity to maintain specific behaviors, despite frustration and fatigue, in anticipation of a delayed reward following prior intermittent reinforcement. High Ps is linked to eagerness, ambitiousness and perfectionism (*Cloninger, 1999*).

Character refers to the higher-order self-regulatory cognitive processes underlying individual differences in goals, motives, values and standards. Character consists of three dimensions that reflect the intrapersonal, interpersonal and transpersonal self (*Cloninger, Svrakic & Przybeck, 1993*). Self-Directedness (SD) refers to the individual differences in the intrapersonal self, who is aware of being an autonomous individual with motives, standards and goals. SD functions as the self-regulatory processes in goal-oriented behaviors, meaning that individuals with high SD are responsible, purposeful, resourceful and self-actualizing. Cooperativeness (C) refers to individual differences in the interpersonal self, who is aware of being a member of a group in which each member has rights and responsibilities. People with high C are tolerant, empathic, helpful, principled and compassionate (*Cloninger, Svrakic & Przybeck, 1993*). Self-Transcendence (ST) refers to individual differences in the transpersonal self, who is aware of being a part of a larger whole, such as nature or the universe. People characterized by high ST are easily absorbed in what they love to do and are altruistic and spiritual (*Cloninger, Svrakic & Przybeck, 1993*).

The psychobiological model of personality was initially developed for clinical assessment in general and clinical populations, but has since proved to be a valid framework for describing normal and abnormal human personality (*Cloninger & Zohar, 2010*). The dimensions of the psychobiological model of personality are causal antecedents of several

indicators of mental health, including psychopathology (both Axis I and Axis II conditions as proposed by the DSM-IV-TR). Recent studies support this model's suitability for describing the personality variables underlying health and all components of wellbeing, as well as ill-being, in adults (*Cloninger & Zohar, 2010*; *Josefsson et al., 2011*) and adolescents (*Moreira et al., 2012*; *Moreira et al., 2015*; *Schütz, Archer & Garcia, 2013*). Thus, this model is a suitable framework from which to investigate the personality characteristics of IPV victims and their associations with psychopathological symptoms.

With the above evidence considered, the aims of the present study were; (a) to identify the unique personality characteristics of women victims of IPV via a comparison with a control group who had not experienced IPV, and (b) to assess whether the experience of IPV moderates the associations between personality characteristics and psychopathological symptoms.

## MATERIALS & METHODS

### Participants

The study used an unmatched case-control design, allowing for the comparison of cases (women who had experienced IPV) and controls (women who had not experienced IPV). Prior to recruitment, we used a freely available online module for calculating the minimum required sample size for unmatched case-control studies (*Sullivan & Soe, 2007*). For this calculation; significance level (alpha) was set at .05; power at 80%; proportion of controls with exposure at 20% (based on *European Union Agency for Fundamental Rights, 2014*); ratio of sample size, Controls/Cases at 1:1, and proportion of cases with exposure at 80%. The results indicated that a minimum of 28 cases (12 cases and 14 controls) was required. Based on these results, and the prevalence of IPV victims seeking support, we determined that 50 cases and 50 matched controls would be an acceptable sample size. A consecutive sampling method was then used to recruit 50 women from specialist projects that provide support to victims of IPV. The recruitment process was conducted by a team of social workers, medical professionals, and psychologists, all of whom worked for these projects. Controls (who were women known to this team from the local community) were selected so that both groups would be approximately similar in terms of age and occupational status, although no specific matching method was used. Women were eligible to be a control if they confirmed that they had never been victims of IPV and had no formal or informal records or suspicions of IPV. All participants were from the north of Portugal where, according to a relatively recent EU-wide survey, 19% of ever-partnered women aged 18–74 years report experiencing IPV at least once in their lifetime (*European Union Agency for Fundamental Rights, 2014*). This study was approved by an Internal Review Board (CIPD-2009-01) at the Research Center for Positive Development (CIPD), Portugal, and only participants who provided written informed consent where included.

### Instruments

#### Personality

We evaluated personality dimensions using the Temperament and Character Inventory–Revised (TCI-R; *Cloninger, 1999*), a self-report instrument with 240 items, each scored

on a 5-point Likert scale from 1 (*definitely false*) to 5 (*definitely true*). This instrument captures the four temperament (NS, HA, RD, and Ps), and three character dimensions (SD, C, and ST) of the psychobiological model of personality. Each of these scales is subdivided into between 3 and 5 subscales that capture different aspects of each personality dimension. The Portuguese version of the TCI-R, which we used in this study, has shown good psychometric properties in the Portuguese population (*Moreira et al., 2017*).

### Psychopathology symptoms

To evaluate psychopathology, we used the Portuguese version of the Brief Symptom Inventory (BSI; *Derogatis & Spencer, 1982*). This instrument consists of 53 items that assess 9 dimensions of psychopathology symptoms. Items are scored using a 5-point Likert scale ranging from 0 (*never*) to 4 (*many times*). The 9 symptom dimensions are: Somatization, Obsession-Compulsion, Interpersonal Sensitivity, Depression, Anxiety, Hostility, Phobic anxiety, Paranoid ideation and Psychoticism. The Portuguese version of the BSI (*Canavarro, 1999*) has been shown to have acceptable psychometric properties, with alpha values that vary between .62 (Psychoticism) and .80 (Somatization).

## Procedures
### Data collection procedures

The psychologists, social workers and medical professionals working for the support projects collaborated in the collection of data for the IPV group. Prior to data collection, the study aims were explained to participants (individually) and informed consents were obtained. Participants in the IPV group completed both questionnaires (paper format) in a technician's office. Participants in the non-IPV completed the questionnaires in an office in the community. For both groups, the order in which we presented the two questionnaires (TCI-R and BSI) was counterbalanced to control for possible order effects. Participants took approximately 1 h to complete the questionnaires.

### Data analysis

We used statistical methods appropriate for non-matched case-control designs given that we did not use a specific matching procedure to pair cases with controls (instead groups were roughly equated based on age and occupational status). First, group differences based on demographic variables were assessed. For categorical demographic variables—civil status (married, single, divorced or other), occupation status, education level (less than secondary school, secondary school or above secondary school), and violence type (psychological, psychological + physical, or psychological + physical + sexual)—group differences were assessed using Cramer's V. Differences in age were assessed using an independent samples $t$-test. Because cases and controls were statistically different for one of these variables (education level), we tested the main effect of group for each personality dimension and psychopathology symptoms using a series of ANCOVAs. For these analyses, education level was included as a covariate. To assess the associations between dimensions of personality and psychopathological symptoms in victims of IPV we calculated Pearson's correlations between TCI-R and BSI subscales. This correlational analysis was only applied to the IPV group ($n = 50$). We interpreted the effect size of correlation coefficients in accordance with
**Table 1 Demographic characteristics of participants.**

| | | IPV (n = 50) | Non-IPV (n = 50) | Total (n = 100) | IPV vs. Non-IPV comparison |
|---|---|---|---|---|---|
| | | M (SE) | M (SE) | M (SE) | |
| | Age | 41.2 (1.85) | 37.4 (1.53) | 39.3 (1.21) | t = 1.56, p = .122 |
| | | % | % | % | |
| | Married | 60 | 56 | 58 | |
| Civil status | Single | 26 | 38 | 32 | Cramer's V = .169, p = .581 |
| | Divorced | 6 | 2 | 4 | |
| | Other | 8 | 4 | 6 | |
| | Homemaker | 22 | 10 | 16 | |
| Occupation status | Student | 2 | 8 | 5 | Cramer's V = .252, p = .095 |
| | Unemployed | 14 | 6 | 10 | |
| | Other occupation | 62 | 76 | 69 | |
| | Less than Secondary School | 84 | 56 | 70 | |
| Education | Secondary School | 16 | 10 | 13 | Cramer's V = .453, p < .001 |
| | Above Secondary School | 0 | 34 | 17 | |
| | Psychological | 20 | – | 10 | |
| Violence type | Psychological + Physical | 62 | – | 31 | – |
| | Psychological + Physical + Sexual | 18 | – | 9 | |

the guidelines offered by *Cohen (1988)*. Finally, we conducted a series of linear multiple regression analyses, using the entire sample ($n = 100$), to assess whether the experience of IPV interacts with personality dimensions in explaining variance in total symptoms. For these analyses, personality dimensions and IPV (coded as a dummy variable) were mean centered.

## RESULTS

### Demographic statistics
Table 1 presents the demographic characteristics of the study participants. Participants in the IPV group had significantly lower education levels than participants in the non-IPV group, with none of the IPV group achieving higher than a secondary-school level of education (Cramer's $V = 0.45$, $p < .001$). There were no significant differences between groups in terms of age, civil status, or occupation status. The majority of the IPV group suffered a combination of physical and psychological violence (62%).

### Differences in personality dimensions
Figure 1 presents personality profiles for the IPV and non-IPV groups. In order to standardize scores across the different personality dimensions, Figure 1 presents *z* scores (representing number of standard deviations from the mean). The IPV group was characterized by elevated Harm Avoidance and Self-Transcendence, and lower Novelty Seeking, Reward Dependence, Persistence, Self-Directedness and Cooperativeness compared to the non-IPV group. Next, we conducted a series of ANCOVAs (based on raw scores) to assess the significance of these observed differences after controlling for

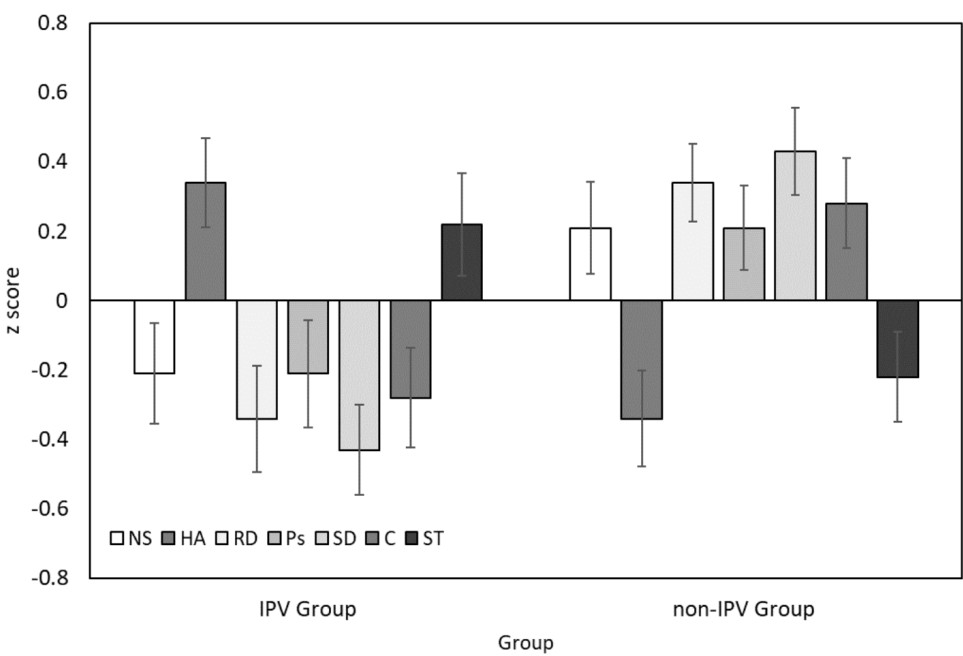

**Figure 1** **Personality profiles for women victims of IPV (IPV Group) and a control group (non-IPVgroup).** Error bars correspond to ±1 standard error. NS, Novelty Seeking; HA, Harm Avoidance; RD, Reward Dependence; Ps, Persistence; SD, Self-Directedness; C, Cooperativeness; ST, Self-Transcendence.

group differences in education (see Table 2). These analyses indicated that the largest observed effect of group was for Self-Directedness: the IPV group had significantly lower Self-Directedness than the non-IPV group ($F = 13.44$, $p < .001$). Women victims of IPV also presented significantly lower levels of Reward Dependence ($F = 7.52$, $p = .007$), and significantly higher levels of Harm Avoidance ($F = 5.52$, $p = .021$) and Self-Transcendence ($F = 7.90$, $p = .006$).

## Differences in psychopathology symptoms

Figure 2 presents the mean summed scores of the BSI symptom subscales for the IPV and non-IPV groups. A series of ANCOVAs (see Table 3) indicated that participants in the IPV group had significantly higher levels of all symptoms. The largest effect of group across all symptoms, after controlling for differences in education, was observed for depression ($F = 45.43$, $p < .001$).

## Correlations between personality dimensions and psychopathological symptoms in victims of IPV

The pattern of correlations across symptom dimensions was broadly consistent for each personality dimension in the IPV group (see Tables 4 and 5). Medium-sized ($|.50| > r > |.30|$; *Cohen, 1988*) correlations were found between Novelty Seeking and psychoticism ($r = -.42$); between Self-Directedness and hostility ($r = -.41$), paranoid ideation ($r = -.45$) and psychoticism ($r = -.50$); and between Self-Transcendence and interpersonal Sensitivity ($r = -.43$). Harm Avoidance was the only temperament

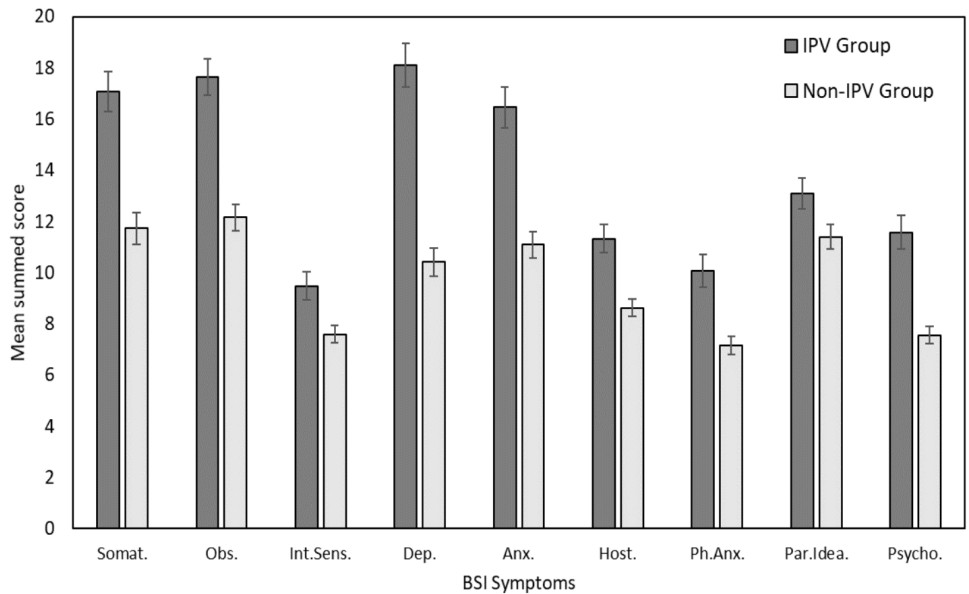

**Figure 2** **Mean average summed scores across BSI symptoms for women victims of IPV (IPV Group) and a control group (Non-IPV Group).** Error bars correspond to ±1 standard error. Somat., Somatization; Obs., Obsession-Compulsion; Int. Sens., Interpersonal Sensitivity; Dep., Depression; Anx., Anxiety; Host., Hostility; Ph.Anx., Phobic Anxiety; Par.Idea., Paranoid Ideology; Psycho., Psychoticism.

**Table 2** **Personality characteristics of women victims of IPV compared to a control group.**

| Personality dimension | IPV group (n = 50) | | Non-IPV group (n = 50) | | ANCOVA: main effect of group controlling for education level | | | | Scale reliability |
|---|---|---|---|---|---|---|---|---|---|
| | M | SE | M | SE | F | p | Partial $\eta^2$ | Power | α |
| Novelty Seeking | 89.16 | 1.73 | 94.16 | 1.56 | 0.86 | .356 | .01 | .15 | .77 |
| Harm Avoidance[*] | 109.32 | 2.02 | 98.80 | 2.15 | 5.52 | .021 | .05 | .64 | .88 |
| Reward Dependence[**] | 97.38 | 1.81 | 105.34 | 1.31 | 7.52 | .007 | .07 | .78 | .78 |
| Persistence | 114.26 | 2.25 | 120.40 | 1.77 | 1.69 | .119 | .02 | .25 | .86 |
| Self-Directedness[***] | 127.14 | 2.66 | 144.62 | 2.58 | 13.44 | <.001 | .12 | .95 | .91 |
| Cooperativeness | 132.82 | 2.04 | 140.66 | 1.85 | 3.53 | .063 | .04 | .46 | .86 |
| Self-Transcendence[**] | 82.70 | 1.61 | 77.86 | 1.42 | 7.90 | .006 | .08 | .80 | .78 |

**Notes.**
Main effect of education level (covariate) not presented in ANCOVA output.
[*]$p < .05$.
[**]$p < .01$.
[***]$p < .001$.

dimension positively correlated with psychopathology symptoms: paranoid ideation ($r = .46$), psychoticism ($r = .43$) and interpersonal sensitivity ($r = .41$).

### The moderating effect of IPV on the association between personality and psychopathological symptoms

A correlational analysis indicated that total symptoms were significantly correlated with education level ($r = -.25$, $p = .011$) and type of IPV ($r = .54$, $p < .001$), and consequently

**Table 3 Psychopathological symptoms in women victims of IPV compared to a control group.**

| Symptoms | IPV group ($n = 50$) | | Non-IPV group ($n = 50$) | | ANCOVA: main effect of group controlling for education level | | | | Scale reliability |
|---|---|---|---|---|---|---|---|---|---|
| | M | SE | M | SE | F | p | Partial $\eta^2$ | Power | $\alpha$ |
| Somatization[***] | 17.06 | 0.78 | 11.74 | 0.62 | 19.53 | <.001 | .17 | .99 | .84 |
| Obsession[***] | 17.64 | 0.73 | 12.16 | 0.51 | 33.38 | <.001 | .26 | 1.00 | .84 |
| Interpersonal sensitivity[**] | 9.48 | 0.54 | 7.60 | 0.33 | 7.81 | .006 | .08 | .79 | .80 |
| Depression[***] | 18.10 | 0.84 | 10.42 | 0.55 | 45.43 | <.001 | .32 | 1.00 | .91 |
| Anxiety[***] | 16.46 | 0.80 | 11.10 | 0.51 | 26.06 | <.001 | .21 | 1.00 | .86 |
| Hostility[***] | 11.32 | 0.55 | 8.62 | 0.34 | 16.64 | <.001 | .15 | .98 | .77 |
| Phobic anxiety[**] | 10.08 | 0.65 | 7.16 | 0.35 | 11.96 | .001 | .11 | .93 | .80 |
| Paranoid ideation[*] | 13.10 | 0.60 | 11.40 | 0.47 | 4.92 | .029 | .05 | .59 | .72 |
| Psychoticism[***] | 11.58 | 0.65 | 7.56 | 0.35 | 29.40 | <.001 | .23 | 1.00 | .80 |

**Notes.**

Main effect of education level (covariate) not presented in ANCOVA output.

[*]$p < .05$.

[**]$p < .01$.

[***]$p < .001$.

these variables were included as covariates in the following regression models. Age was not significantly correlated with total symptoms in our sample ($r = .03$), although we also included this variable as a covariate given that prior research has shown clear age effects on psychopathology (e.g., *Erskine et al., 2007*).

Table 6 presents the results of seven independent regression analyses. After controlling for the effect of IPV, education level, age, and violence type, Harm Avoidance ($b = .75$, $p < .001$) and Self-Directedness ($b = -.67$, $p < .001$) were significant predictors of total symptoms in their respective models. Crucially, while Novelty Seeking was not a significant predictor of symptoms, it was found to have a significant interaction with IPV ($b = -1.10$, $p = .034$). This interaction is plotted in Fig. 3. (Note that we used z scores for Novelty Seeking and raw scores for total symptoms to graph this significant interaction). The simple slopes for this interaction indicated that there was a non-significant positive relationship between Novelty Seeking and total symptoms in the non-IPV controls, $b = .25$, 95% CI [−0.51, 1.00], and a significant negative relationship between Novelty Seeking and total symptoms in victims of IPV, $b = -.85$, 95% CI [−1.53, −0.17]. IPV did not interact with the remaining six personality dimensions in the prediction of psychopathology symptoms.

## DISCUSSION

The principle finding of this study was that women victims of IPV had a significantly different character and temperament profile to a group of control women who had reported never experiencing IPV. The results also show that personality dimensions were differentially associated with psychopathology symptoms in victims of IPV, and that the experience of IPV moderated the association between Novelty Seeking and total symptoms. The following discussion will address these findings in turn.

Moreira et al. (2019), *PeerJ*, DOI 10.7717/peerj.6647

**Table 4  Correlations between temperament dimensions and psychopathological symptoms in women victims of IPV ($n = 50$).**

| Temperament | | Psychopathological symptoms | | | | | | | | |
|---|---|---|---|---|---|---|---|---|---|---|
| Subscales | Scales | Somatization | Obsession | Interpersonal sensitivity | Depression | Anxiety | Hostility | Phobic anxiety | Paranoid ideation | Psychoticism |
| | **Novelty seeking** | −.070 | **−.344**\* | −.244 | **−.368**\* | −.173 | **−.307**\* | −.055 | −.254 | **−.422**\* |
| NS1 | Exploratory excitability | −.161 | **−.317**\* | −.220 | **−.356**\* | −.192 | **−.372**\* | −.191 | **−.310**\* | **−.389**\* |
| NS2 | Impulsiveness | −.029 | −.223 | −.169 | −.252 | −.034 | −.198 | .006 | −.089 | −.240 |
| NS3 | Extravagance | −.024 | −.216 | −.108 | **−.322**\* | −.035 | −.250 | .033 | −.182 | **−.357**\* |
| NS4 | Disorderliness | .053 | −.202 | −.196 | −.074 | −.197 | .007 | .047 | −.072 | −.158 |
| | **Harm avoidance** | .199 | **.380**\* | **.410**\* | **.302**\* | **.328**\* | .269 | .176 | **.463**\* | **.425**\* |
| HA1 | Anticipatory worry | .201 | **.375**\* | **.431**\* | **.332**\* | **.329**\* | **.373**\* | .101 | **.535**\* | **.478**\* |
| HA2 | Fear of uncertainty | .096 | .272 | .094 | .151 | .130 | .086 | .202 | .081 | .187 |
| HA3 | Shyness | −.005 | .088 | **.319**\* | .074 | .171 | .127 | −.017 | **.352**\* | .166 |
| HA4 | Fatigability | **.331**\* | **.453**\* | **.378**\* | **.366**\* | **.371**\* | .188 | **.312**\* | **.386**\* | **.449**\* |
| | **Reward dependence** | −.139 | −.227 | −.144 | −.131 | **−.331**\* | **−.315**\* | −.150 | **−.389**\* | −.240 |
| RD1 | Sentimentality | −.027 | .065 | .085 | .050 | −.047 | −.054 | .073 | −.043 | .035 |
| RD2 | Openness to communication | −.107 | −.255 | **−.319**\* | −.141 | **−.292**\* | −.247 | −.164 | **−.374**\* | −.252 |
| RD3 | Attachment | −.228 | **−.393**\* | **−.440**\* | **−.339**\* | **−.418**\* | **−.400**\* | −.177 | **−.474**\* | **−.448**\* |
| RD4 | Dependence | −.010 | .050 | **−.281**\* | .136 | −.084 | −.115 | −.099 | −.084 | .097 |
| | **Persistence** | −.180 | −.269 | −.168 | −.211 | −.232 | −.111 | −.060 | **−.358**\* | **−.287**\* |
| PS1 | Resistance to stress | −.195 | **−.352**\* | −.202 | −.243 | −.210 | −.161 | −.200 | −.243 | −.277 |
| PS2 | Work | −.164 | −.187 | −.192 | −.120 | −.172 | −.099 | −.031 | **−.313**\* | −.277 |
| PS3 | Ambition | −.059 | −.135 | −.168 | −.127 | −.136 | −.102 | .076 | **−.366**\* | −.232 |
| PS4 | Perfectionism | −.245 | −.266 | −.205 | −.262 | **−.330**\* | .023 | −.066 | **−.301**\* | −.187 |

**Notes.**
\*$p < 0.05$.
Values in bold reflect a medium association, $r > |.30|$.

**Table 5  Correlations between character dimensions and psychopathological symptoms in women victims of IPV ($n = 50$).**

| Character | | Psychopathological symptoms | | | | | | | | |
|---|---|---|---|---|---|---|---|---|---|---|
| Subscales | Scales | Somatization | Obsession | Interpersonal sensitivity | Depression | Anxiety | Hostility | Phobic anxiety | Paranoid ideation | Psychoticism |
| | **Self-directedness** | −.196 | **−.377**[*] | −.037 | **−.380**[*] | −.220 | **−.410**[*] | −.158 | **−.446**[*] | **−.500**[*] |
| SD1 | Responsibility | −.198 | **−.456**[*] | **−.361**[*] | **−.411**[*] | −.258 | **−.365**[*] | −.196 | **−.426**[*] | **−.533**[*] |
| SD2 | Purposeful | −.162 | **−.390**[*] | **−.295**[*] | **−.417**[*] | −.265 | **−.333**[*] | −.170 | **−.368**[*] | **−.499**[*] |
| SD3 | Resourcefulness | −.157 | **−.361**[*] | **−.302**[*] | **−.354**[*] | −.162 | −.208 | −.164 | **−.432**[*] | **−.345**[*] |
| SD4 | Self-acceptance | −.061 | −.005 | −.230 | −.091 | −.030 | **−.383**[*] | .116 | **−.307**[*] | −.185 |
| SD5 | Congruency | −.160 | −.186 | **−.420**[*] | −.171 | −.089 | −.270 | −.150 | −.220 | **−.295**[*] |
| | **Cooperativeness** | −.191 | −.061 | −.125 | −.099 | −.135 | **−.301**[*] | −.096 | **−.324**[*] | −.149 |
| CO1 | Social acceptance | −.136 | −.015 | −.257 | .036 | −.076 | −.263 | −.149 | −.084 | .000 |
| CO2 | Empathy | −.186 | −.148 | .242 | −.154 | −.201 | −.176 | −.005 | **−.378**[*] | −.223 |
| CO3 | Helpfulness | −.100 | −.059 | −.266 | −.055 | −.178 | −.188 | −.003 | **−.395**[*] | −.161 |
| CO4 | Compassion | −.228 | −.145 | −.040 | −.239 | −.189 | **−.429**[*] | −.151 | **−.383**[*] | **−.316**[*] |
| CO5 | Conscience | −.096 | .094 | −.201 | .011 | .066 | −.075 | −.015 | −.110 | .075 |
| | **Self-transcendence** | −.126 | −.172 | **−.430**[*] | −.161 | −.266 | −.080 | −.009 | **−.366**[*] | −.253 |
| ST1 | Self-forgetful | −.091 | −.258 | **−.313**[*] | −.196 | −.160 | −.050 | −.073 | −.184 | −.254 |
| ST2 | Transpersonal identification | −.079 | −.136 | −.134 | −.119 | −.150 | −.174 | .115 | **−.419**[*] | **−.298**[*] |
| ST3 | Spiritual acceptance | −.092 | .053 | −.264 | −.010 | −.248 | .046 | −.041 | −.188 | .024 |

**Notes.**

[*]$p < 0.05$.

Values in bold reflect a medium association, $r > |.30|$.

**Table 6** Summary of seven independent regression analyses testing predictors of total symptoms ($n =$ 100).

| Model | Predictors | b | p | Model summary | | | |
|---|---|---|---|---|---|---|---|
| | | | | $R^2$ | F | p | Power |
| NS | NS | −0.30 | .245 | .36 | 8.62 | <.001 | 1.00 |
| | IPV | 17.07 | .251 | | | | |
| | NS × IPV* | −1.10 | .034 | | | | |
| | Age | −0.07 | .797 | | | | |
| | Education | −1.93 | .668 | | | | |
| | Violence type | 4.4.3 | .185 | | | | |
| HA | HA* | 0.75 | <.001 | .41 | 10.82 | <.001 | 1.00 |
| | IPV | 13.86 | .330 | | | | |
| | HA × IPV | 0.46 | .225 | | | | |
| | Age | −0.13 | .960 | | | | |
| | Education | 0.93 | .833 | | | | |
| | Violence type | 4.02 | .208 | | | | |
| RD | RD | −0.28 | .318 | .35 | 8.50 | <.001 | 1.00 |
| | IPV | 15.53 | .302 | | | | |
| | RD × IPV | −1.03 | .064 | | | | |
| | Age | −0.06 | .819 | | | | |
| | Education | −3.00 | .504 | | | | |
| | Violence type | 4.47 | .182 | | | | |
| Ps | Ps | −0.30 | .152 | .35 | 8.21 | <.001 | 1.00 |
| | IPV | 15.57 | .300 | | | | |
| | Ps × IPV | −0.55 | .197 | | | | |
| | Age | −0.13 | .631 | | | | |
| | Education | −2.85 | .528 | | | | |
| | Violence type | 4.63 | .170 | | | | |
| SD | SD* | −0.67 | <.001 | .44 | 12.14 | <.001 | 1.00 |
| | IPV | 5.96 | .673 | | | | |
| | SD × IPV | −0.25 | .95 | | | | |
| | Age | 0.02 | .943 | | | | |
| | Education | 0.37 | .930 | | | | |
| | Violence type | 4.93 | .116 | | | | |
| C | C | −0.29 | .190 | .33 | 7.78 | <.001 | 1.00 |
| | IPV | 13.69 | .373 | | | | |
| | C × IPV | −0.45 | .303 | | | | |
| | Age | −0.11 | .694 | | | | |
| | Education | −2.95 | .529 | | | | |
| | Violence type | 4.96 | .148 | | | | |

*(continued on next page)*

**Table 6** (*continued*)

| Model | Predictors | *b* | *p* | Model summary | | | |
|---|---|---|---|---|---|---|---|
| | | | | *R²* | *F* | *p* | *Power* |
| | ST | −0.27 | .336 | | | | |
| | IPV | 21.45 | .160 | | | | |
| ST | ST × IPV | −0.83 | .134 | .34 | 7.90 | <.001 | 1.00 |
| | Age | −0.21 | .439 | | | | |
| | Education | −3.45 | .449 | | | | |
| | Violence type | 3.93 | .247 | | | | |

**Notes.**

Constant not included in model summaries.

NS, Novelty seeking; HA, Harm avoidance; RD, Reward dependence; Ps, Persistence; SD, Self-Directedness; CO, Cooperativeness; ST, Self-transcendence; IPV, Intimate Partner Violence.

*$p < .05$.

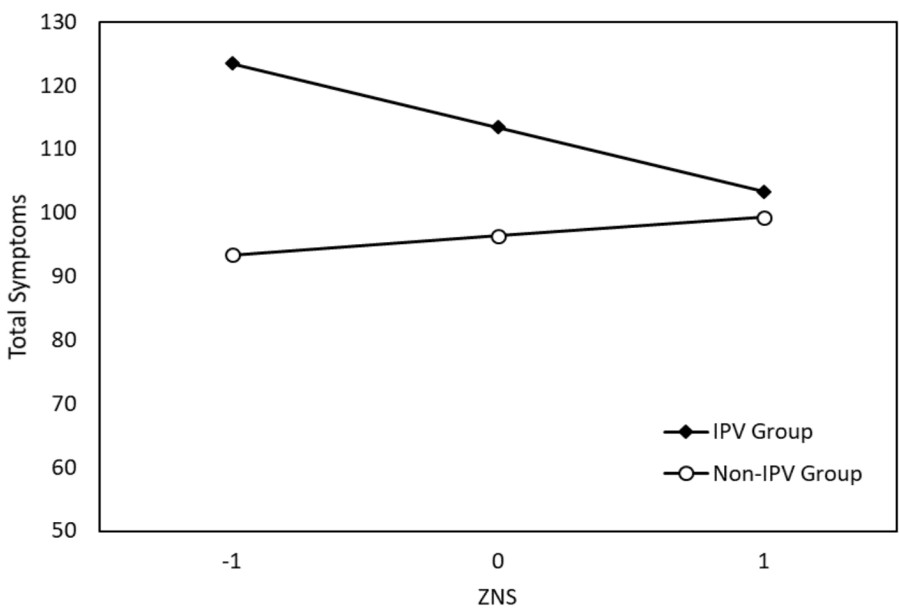

**Figure 3** Significant interaction effect of Novelty Seeking (NS) and IPV on total symptoms.

## IPV and personality characteristics

The IPV group had a character configuration of low Self-Directedness, low Cooperativeness, and high Self-Transcendence (a *scT* configuration). In terms of temperament, these women had high levels of Harm Avoidance, and low scores in Novelty Seeking, Reward Dependence and Persistence: an *nHrp* configuration. These personality configurations (*Cloninger, Svrakic & Przybeck, 1993*) imply that these women were more likely to be immature, disorganized, and schizotypal (associated with *scT* configuration), and to have weak social skills and avoidant behavior (associated with *nHrp* configuration). These character and temperament features are typical of psychological organizations associated with schizotypal personality disorder (*Cloninger, Bayon & Svrakic, 1998*; *Svrakic et al.,*

*2002*), one of the most dominant personality disorders in this population (*Pérez-Testor et al., 2007*; *Pico-Alfonso, Echeburúa & Martinez, 2008*).

If the personality characteristics associated with the IPV group are considered as antecedent causes of individual differences (indeed studies have demonstrated personality dimensions have predictive validity in prospective studies; *Grucza & Goldberg, 2007*), our results imply that women inherently more tolerant of routine and conformity, and less willing to explore new situations are more vulnerable to IPV. Low levels of Novelty Seeking may account for a higher tolerance to repeated patterns of abusive interactions and may partially account for the difficulties many women have in leaving abusive relationships (*Cloninger, Svrakic & Przybeck, 1993*). The IPV group also showed high levels of Harm Avoidance—indeed significantly higher Harm Avoidance than the non-IPV group—a trait associated with high levels of negative affectivity (*Cloninger, Svrakic & Przybeck, 1993*). Negative emotions have been shown to interfere with executive functioning (*Ochsner et al., 2004*) and this may influence the ability of IPV victims to conceive of, plan, and then actively seek new experiences (i.e., new relationships). The significantly lower levels of Reward Dependence found in the IPV group compared to the non-IPV group also imply that these women are dispositionally more tolerant of the lack of warm and secure emotional reward from abusive relationships. These findings are consistent with past evidence about the indicators of the biopsychosocial functioning of victims. Neuroticism, as defined by the Five-Factor model of personality, corresponds to high Harm Avoidance and low Self-Directedness in the psychobiological model (*Cloninger, 2006*). This pattern of high Harm Avoidance and low Self-Directedness was characteristic of the IPV group in our study. Neuroticism (both in victims and aggressors) is as a major predictor of starting/precipitating the aggressive situations of IPV (*Hellmuth & McNulty, 2008*).

Thus far, we have discussed the present results in terms of how personality may increase vulnerability to IPV. As with all correlational data, our results might equally demonstrate the association between IPV and stress-related personality changes. In support of this proposal, longitudinal research has shown that the experience of traumatic events in adulthood can cause changes to personality (*Josefsson et al., 2013*) including increases in neuroticism (high Harm Avoidance and low Self-Directedness) and decreases in agreeableness (*Löckenhoff et al., 2009*). Consequently, some care should be taken before drawing the conclusion that individuals with certain personality profiles are at increased risk of suffering IPV.

## Personality and psychopathology

Personality influences the nature, intensity and frequency of exposure to stress factors. This interferes with the coping strategies used by individuals in stressful situations (*Connor-Smith & Flachsbart, 2007*). The selection and use of coping strategies has been shown to be correlated with several Big Five personality dimensions, and in particular, neuroticism has been linked to less adaptive strategies (*Connor-Smith & Flachsbart, 2007*). The low Novelty Seeking, low Reward Dependence, low Persistence, low Self-Directedness and high Harm Avoidance found in our IPV group (which corresponds closely to neuroticism) therefore implies that victims may not use adequate or adaptive problem-solving and cognitive restructuring strategies (*Connor-Smith & Flachsbart, 2007*). The elevation of
Self-Transcendence found in victims of IPV is also likely to be related to the coping skills necessary for dealing with daily suffering (*Coward & Reed, 1996*).

Past studies have demonstrated that personality has an influence on psychopathology via its effect on coping strategies (*Auerbach et al., 2010*), a finding which is consistent with the elevation of psychopathology in our IPV group. High Harm Avoidance is a strong predictor of the presence and severity of anxiety and mood disorders (*Cloninger & Zohar, 2010*; *Gonçalves & Cloninger, 2010*) because it potentiates the impact of negative life events and hinders the development of adaptive coping strategies (*Hofmann & Bitran, 2007*). Individuals with high Harm Avoidance tend to be highly sensitivity to negative life events and have difficulty dealing with their impact (*Cole, Logan & Shannon, 2005*). Such characteristics may contribute to the high prevalence of psychopathology in victims of IPV (*Howard et al., 2010*; *Nurius & Macy, 2010*; *Okuda et al., 2011*) and their difficulty in stopping abusive relationships (*Palker-Correll & Marcus, 2004*; *Renner & Slack, 2006*). In addition, the character dimensions Self-Directedness and Cooperativeness (which were low in our IPV sample) are predictors of a range of positive indicators of health, including subjective wellbeing (*Cloninger & Zohar, 2010*; *Josefsson et al., 2013*). Such associations may help explain the low levels of satisfaction in life (*Dyrenforth et al., 2010*), low quality of life (*Adeodato et al., 2005*), and the high psychological distress and suicidality (*Leiner et al., 2008*) among victims of IPV. Finally, personality dimensions have been shown to predict the development of severe conditions after trauma (such as PTSD; *North, Abbacchi & Cloninger, 2012*), to mediate the relationship between depression and PTSD (*Bargai, Ben-Shakhar & Shalev, 2007*), and to predict recovery from PTSD in victims (*Kunst, 2010*). High levels of Self-Directedness and Cooperativeness are a protective factor against the development of PTSD (*North, Abbacchi & Cloninger, 2012*). Moreover, a positive affect type of personality (low Harm Avoidance and high Self-Directedness) predicts recovery from PTSD among victims.

The final major finding of the present study, in addition to highlighting the important role of Harm Avoidance and Self-Directedness for understanding psychopathological symptoms, was the significant interaction between Novelty Seeking and IPV. Specifically, high Novelty Seeking in women victims of IPV was linked to fewer total symptoms, while high Novelty Seeking in the control group was related to elevated total symptoms. Novelty Seeking thus appears to plays an important role in the expression of individuals' experiences of IPV via the manifestation of psychopathological symptoms. It is important to highlight that symptoms from affective domains (depression and anxiety), elsewhere referred to as internalizing symptoms, are a prevalent feature in individuals with elevated Harm Avoidance, such is the case in women victims of IPV. High Novelty Seeking refers to a behavioral phenotype of sensitivity to novelty that expresses as higher openness to new experiences and a tendency to externalize internal states. High sensitivity to novelty, and in particularly higher openness to new experience, implies that women with high Novelty Seeking express an intolerance for monotony and may be more easily provoked to a fight or flight response (*Cloninger, 1987*). Such women may have a stronger drive to explore alternatives to their abusive relationships. Individuals high in Novelty Seeking are also more likely to interact with their environments by externalizing their subjective experiences (both

positive and negative), and this can manifest as impulsive overt expressions of feelings, thoughts and internal states. These impulsive overt expressions can serve as important signals about an individual's subjective experience, and can function as a boundary setter and/or a sign of empowerment within an abusive relationship. Moreover, overt expressions of internal states to individuals outside of the relationship can be an important coping mechanism.

Past research has shown that effective interventions for IPV victims promote the development of internal resources and strengths, particularly personal agency mechanisms and socio-cognitive processes (e.g., *Burnette & Cannon, 2014*; *Hansen, Eriksen & Elklit, 2014*; *Hoyeck et al., 2014*). Results from studies on change processes (both spontaneous and intervention-promoted) in IPV victims converge on the finding that the psychobiological processes associated with change are well captured by personality dimensions. Processes such as self-esteem and self-efficacy, reflection, awareness, meaning making, or self-identity are components of higher-order dimensions such the individuals' systems of beliefs, standards and goals. These in turn refer to crucial dimensions of personality, as is revealed by their inclusion in many theories of personality. For example, an individual's systems of beliefs, standard and goals refer to the dimensions of character in Cloninger's psychobiological model of personality (*Cloninger, Svrakic & Przybeck, 1993*).

## Limitations

Some limitations of the present study require consideration. Firstly, we used a cross-sectional design and were therefore unable to draw any conclusions about how personality might change over time because of IPV. Further, the way we analyzed the associations between personality dimensions and psychopathology (correlational) means that the reader should be wary of inferring causality. Future investigations should use appropriate regression analyses and longitudinal designs if aiming to understand whether personality characteristics serve as antecedents to psychopathology. It should be noted that for such analyses a larger sample size than the one used in this study would be required to achieve sufficient power.

Another limitation to consider is that victims of IPV were recruited from specialist projects. The implication of this aspect of the design is that our sample is unlikely to be fully representative of all women victims of IPV. Women seeking support from specialist projects are likely to represent only the most serious experiences of IPV, and will not include the large number of women who conceal their victimization. Furthermore, women who do reveal their experiences of IPV and seek or accept help from such projects may have specific personality profiles that are not representative of all victims. Consequently, the personality profile for IPV victims identified in our study may correspond to a specific subsample of women victims. Future studies with alternative sampling techniques will be required to verify our findings.

A final limitation of this study is that we did not collect data for variables that might mediate the relationship between high Harm Avoidance, low Self-Directedness and involvement in violence, and thus did not control for them. Future studies should therefore

control variables such as alcohol abuse (*Egan & Hamilton, 2008*), effective problem-solving skills, levels of experienced stress (*Hellmuth & McNulty, 2008*), satisfaction with life (*Dyrenforth et al., 2010*), and history of maltreatment in childhood or characteristics of abuse (*Bensley, Van Eenwyk & Simmons, 2003*).

## CONCLUSIONS

Our study highlights the relevance of personality dimensions for understanding the experience of being a victim of IPV. Harm Avoidance and Self-Directedness are of particular importance for understanding an individual's risk for experiencing psychopathological symptoms. Novelty Seeking appears to play an important role in the expression of individuals' experiences of IPV. These results have important implications for research and practice (especially for the development and implementation of interventions). Specifically, they suggest that the personality of each IPV victim needs to be considered when planning interventions and practices, from crisis interventions to medium- and long-term interventions. Indeed, this corresponds to a major challenge in current mental health services: shifting from an emphasis on symptoms to a person-centered approach to individuals.

### Funding

This work is financed by national funds by FCT—Foundation for Science and Technology, under the Project UID/PSI/04375/2016. The funders had no role in study design, data collection and analysis, decision to publish, or preparation of the manuscript.

### Grant Disclosures

The following grant information was disclosed by the authors:
FCT—Foundation for Science and Technology: Project UID/PSI/04375/2016.

### Competing Interests

C. Robert Cloninger is an Academic Editor for PeerJ.

### Author Contributions

- Paulo A.S. Moreira conceived and designed the experiments, analyzed the data, contributed reagents/materials/analysis tools, prepared figures and/or tables, authored or reviewed drafts of the paper, approved the final draft.
- Márcia Pinto performed the experiments, contributed reagents/materials/analysis tools, prepared figures and/or tables.
- C. Robert Cloninger contributed reagents/materials/analysis tools, authored or reviewed drafts of the paper, approved the final draft.
- Daniela Rodrigues performed the experiments.
- Carlos Fernandes da Silva conceived and designed the experiments, authored or reviewed drafts of the paper, approved the final draft.

## Human Ethics

The following information was supplied relating to ethical approvals (i.e., approving body and any reference numbers):

The Director Board of the Centro de Investigação em Psicologia para o Desenvolvimento (CIPD) approved this research (CIPD-2009-01).

## Data Availability

Data is available at Open Science Framework: https://osf.io/3y2tk.

## Supplemental Information

Supplemental information for this article can be found online at http://dx.doi.org/10.7717/peerj.6647#supplemental-information.

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
