# Peer review of "Understanding the experience of psychopathology after intimate partner violence: the role of personality"

_PeerJ, doi:10.7717/peerj.6647_

## Round 0.1 · original submission · Major Revisions

This is a simple case-control study, which should be first specified at the beginning of the methodology section. Further details on the design of case-control study should be provided, including study type (i.e., matched or unmatched), and procedures of control selection. The study recruited controls from communities, not institutions, but the IPV subjects are recruited from shelters. The authors also need to consider the limitation of unmatched study settings.

·

Basic reporting

Some sentences need to be written more clearly, for example, In line 133-136, page 9.

Experimental design

1. In line 190, page 10. The sample size is 50 for IPV group and 50 for control. Do you have a sample size calculation? Or you should emphasize it on the limitation.

2. In line 192, page 10. Women in the IPV group were recruited from projects whose main goal was to support victims of intimate violence. I am concerning the representative of the sample, how about these women were recruited? One of the most possible reason, maybe these women have psychopathological symptoms and join the projects looking for help. How about other women who were victims without looking for help?

Validity of the findings

In line 242 of page 12, more detail of the demographic variable should be given, you’d better list in a table.

Reviewer 2 ·

Basic reporting

This is an interesting study focusing on the association between personality and psychopathological symptoms in women victims of intimate partner violence.
a. The writing needs further revision for English language. For example, on Line 183, there are two aims of this study, and it should be “the aims of the present article were…”.
b. In the introduction on Line 59 and 66, there should be citations for the prevalence of IPV in Western Europe and the Cloninger’s model.
c. Because the participants were recruited in Portugal, what is the prevalence of IPV against women there, including physical, psychological and sexual violence?
d. The structure conforms to PeerJ standards.
e. The authors mentioned there are figures in the current manuscript, but there were not figures.
f. All the titles for tables should be more concise. In tables, “NS”,”HA”,”PS” and so on should use their full spells, or list the full spell under the table. Meanwhile, the authors did not explain what NS1, HA1, PS1 and etc. stood for. If these meant different levels of each dimensions, the explanation should be showed in method section.
g. The raw data is supplied.

Experimental design

a. The study fits the Scope of the journal.
b. The research question is defined.
However, I am concerned that will the victimization have direct or indirect impact on personality and interfere with the association between personality and psychopathological symptoms?
The research questions identified knowledge gap.
c. The survey was conducted following an ethical standard.
d. Methods described are sufficient. But here are my comments on the methods as following:
 There are no clear inclusion and exclusion criteria for both IPV victims and the comparison group.
 How did the authors determine the sample size for victims and non-victims?
 If victims and non-victims were paired to have the same sample size, what was the pairing process?
 The authors stated in the manuscript that they did not randomly recruit participants for control group. This could interfere the validity of results.
 The alpha values of TCI-R and BSI in this study should be reported.
 On line 227 to 229, the authors stated that they randomly selected half of participants to complete TCI-R first. Please explain the reason.

Validity of the findings

a. Line 243 to 249, the authors described the sociodemographic information for both groups. However, it was only stated as “of similar age, occupational and educational status” without providing any statistical values to support the conclusion.
The distribution of marriage status, education, occupation should be compared between two groups.

I suggest the authors make a table for the sociodemographic information.
b. Line 251 to line 259:
 There is no Figure 1 in the manuscript.
 In Table 1, the author did not explain the meanings of each subgroup under NS, HA, RD, PS, SD, CO and ST, nor discuss the differences between victims and non-victims, most of which showed significant differences with p value under 0.05.
c. Line 261 to 266:
 There is no Figure 2.
 In Table 2, if the authors decided to show t value, please keep all values in two decimals.
d. Line 275 to Line 284:
The authors applied dependent regressions to examine the association between personality and psychopathological symptoms. However, the analysis missed several important factors:
 First of all, it should be more clearer according to Line 238 to 240 whether the regression only included victims.
 In the regression analysis, the authors did not consider the impact of sociodemographic factors including age, education, occupation, marriage, etc.
 The analysis did not consider the interactions between different personalities on symptoms.

·

Basic reporting

Moreira et al. conducted a cross-sectional study to evaluate the differences in psychopathology symptoms between women victims and non-victims of IPV, and explore the associations between personality and psychopathological symptoms in women victims of IPV. Generally speaking, the manuscript is clearly presented and adds some new knowledge to existing literature, in my view.

Experimental design

Cross-sectional design.

Validity of the findings

The authors should report the power of the main analyses to support their conclusion.

Additional comments

1. Were the women victims of IPV consecutively recruited from the projects?
2. The power of main analysis should be given.
3. You have performed many t-tests and correction analyses. I am wondering under what circumstances it is more appropriate to use a smaller alpha value instead of the standard 0.05.

---

## Round 0.2 · Major Revisions

Although one of the three reviewers of the 1st round peer-review suggests to accept your revised paper, I am still not satisfied with this revised version. Please carefully revised it according to the below comments and the comments provided by the reviewers.

1. Language of the paper still has some problems, for example, line 97-98, line 217 “nine”. Some sentences are not well-organized, for example, line 103-104 is not related to “personality influences the IPV-psychological symptom relationship. I would suggest the authors to carefully edit the paper again.

2. Descriptions on sample size estimation of this case-control study is still not detailed enough. For example, parameters and their values used should be specified, rather that just to say “50 cases and 50 controls is acceptable”. Please provide details.

3. It is unusual to match cases and controls on occupation. Please specify your considerations.

4. Statistics. Based on the details on the design of this study, this study is a 1:1 case-control study, so t-test used in Table 2 and Table 3 should be paired-t test, but it seems independent-samples t-test was used here.

5. Line 239, I did not find Cohen 1988 on the reference list. Also, I did not find corresponding explanations of correlation coefficients in the section of Results.

6. Line 242-244, it is incorrect to calculate prior power, power can only be calculated post-analysis, or after the study was done. Please delete this sentence, and provide your considerations for sample size estimation.

7. Because IPV and non-IPV groups are not comparable in terms of education. All the analyses comparing the two groups should control for education, however, such analysis was not done. Results are still questionable.

8. Similar distributions of age, marital status, and occupation between IPV and non-IPV groups are not the reason for excluding these variables from the regression analysis, because the dependent variable is symptoms, not IPV!!!

9. Reference list should be carefully checked, for example, some have DOI links, but some not.

10. Figure 1, how Z-scores are generated should be specified in the main text. Further what NS, HA and other abbreviations represent should be specified.

11. Figure 2 has similar issues.

·

Basic reporting

Clear

Experimental design

Meet the standards

Validity of the findings

No comment

·

Basic reporting

no comment

Experimental design

no comment

Validity of the findings

no comment

Additional comments

1. The power analysis is still unclear. You should provide the parameters which were used in the power analysis. Then, the reviewers can repeat your analysis.
2. In Tables 2 to 3, it seems that TCI and BSI scores between IPV and on-IPV are compared by independent-samples t-test but the study design is 1:1 matched case-control according to the descriptions on the methodology of this study. In this case, paried-samples t test should be used.

---

## Round 0.3 · Minor Revisions

This revised paper seems much better than the previous versions. Thank you for your careful revisions. Would you please kindly check it one last time for typos and missing information? For example, I noticed that line 75 says "Golding (199)" which is clearly missing a digit.

I assume this will be an easy task and then I anticipate the article being promptly accepted.

---

## Round 0.4 · Minor Revisions

Thank you for your careful revisions on the submitted work. Before the paper is formally accepted, there is still a subtle concern regarding the title of the current paper.

Because PeerJ is a journal with a wide general audience, lay people might interpret "personality of women victims" (when read in isolation) as placing blame on the women for the violence done to them. So if possible would you please consider revising the title to avoid the possible appearance of victim shaming?

---

## Round 0.5 · accepted · Accept

Thanks for your revisions. The new title seems good.

#